# The Impact of Commercially Available Dry Mouth Products on the Corrosion Resistance of Common Dental Alloys

**DOI:** 10.3390/ma16114195

**Published:** 2023-06-05

**Authors:** Anna Yu. Turkina, Irina M. Makeeva, Oleg N. Dubinin, Julia V. Bondareva, Daniil A. Chernodoubov, Anastasia A. Shibalova, Alina V. Arzukanyan, Artem A. Antoshin, Peter S. Timashev, Stanislav A. Evlashin

**Affiliations:** 1Therapeutic Dentistry Department, Institute for Dentistry, I.M. Sechenov First Moscow State Medical University, 8-2 Trubetskaya Str., 119991 Moscow, Russia; turkina_a_yu@staff.sechenov.ru (A.Y.T.); makeeva_i_m@staff.sechenov.ru (I.M.M.); arzukanyan_a_v@staff.sechenov.ru (A.V.A.); 2Center for Materials Technologies, Skolkovo Institute of Science and Technology, 30, Bld. 1 Bolshoy Boulevard, 121205 Moscow, Russia; o.dubinin@skoltech.ru (O.N.D.); s.evlashin@skoltech.ru (S.A.E.); 3World-Class Research Center, Saint Petersburg State Marine Technical University, Lotsmanskaya Str. 3, 190121 Saint Petersburg, Russia; 4National Research Center “Kurchatov Institute”, 123182 Moscow, Russia; danny.a.tch@gmail.com; 5Institute of Nanotechnology of Microelectronics of the Russian Academy of Sciences, Leninsky Prospect, 32A, 119991 Moscow, Russia; nastia0694@gmail.com; 6Institute for Regenerative Medicine, I.M. Sechenov First Moscow State Medical University, 8-2 Trubetskaya Str., 119991 Moscow, Russia; antoshin_a_a@staff.sechenov.ru (A.A.A.); timashev.peter@gmail.com (P.S.T.); 7World-Class Research Center “Digital Biodesign and Personalized Healthcare”, I.M. Sechenov First Moscow State Medical University, 8-2 Trubetskaya Str., 119991 Moscow, Russia; 8Chemistry Department, Lomonosov Moscow State University, 119234 Moscow, Russia

**Keywords:** Ti-6Al-4V, CoCr, implants, contact pairs, artificial saliva, corrosion resistant

## Abstract

Dental implants are thought to be implanted for life, but throughout their lifespan, they function in aggressive oral environment, resulting in corrosion of the material itself as well as possible inflammation of adjacent tissues. Therefore, materials and oral products for people with metallic intraoral appliances must be chosen carefully. The purpose of this study was to investigate the corrosion behavior of common titanium and cobalt–chromium alloys in interaction with various dry mouth products using electrochemical impedance spectroscopy (EIS). The study showed that different dry mouth products lead to different open circuit potentials, corrosion voltages, and currents. The corrosion potentials of Ti64 and CoCr ranged from −0.3 to 0 V and −0.67 to 0.7 V, respectively. In contrast to titanium, pitting corrosion was observed for the cobalt–chromium alloy, leading to the release of Co and Cr ions. Based on the results, it can be argued that the commercially available dry mouth remedies are more favorable for dental alloys in terms of corrosion compared to Fusayama Meyer’s artificial saliva. Thus, to prevent undesirable interactions, the individual characteristics of not only the composition of each patient’s tooth and jaw structure, but also the materials already used in their oral cavity and oral hygiene products, must be taken into account.

## 1. Introduction

The number of edentulous patients increases every year due to age-related changes, poor hygiene, and bad habits [1]. Over the past decades, dental implants and crowns have been forecasted to become the most common and popular method of missing teeth replacement [2,3]. Other common methods of prosthodontic rehabilitation are removable dentures [3] and zirconia- or CoCr-based metal-ceramic bridges [4]. Despite the multiple clinical benefits, dental implants and other metallic oral appliances may have negative effects on the oral tissues due to corrosion and galvanic interactions between different dental alloys in the saliva [5]. 

Titanium implants are popular in dentistry and commonly coupled with CoCr- or NiCr-based suprastructures [6,7] to improve physicochemical properties or facilitate casting. However, the presence of additional metal inclusions and the constant interaction of metallic intraoral devices with the oral environment can lead to the development of side effects such as mouth-burning syndrome, oral lesions, and systemic reactions to metal ions [5]. Potential side effects are based on the following mechanisms: (1) negative effects of metal ions released as a result of corrosion [8,9] and (2) direct current occurring in case of galvanic coupling of different alloys in saliva [10]. 

Corrosion of dental alloys in the oral cavity is a complex process affected by multiple factors, including saliva properties, oral biofilm, and oral hygiene products used [11]. The strength of the galvanic current and the activity of corrosion in the oral cavity depends primarily on the electrochemical properties of the saliva [12]. The electroconductivity of the mixed saliva varies from 3.5 mS/cm to 4.73 mS/cm [13,14,15]. The decrease in salivary pH leads to an increase in galvanic corrosion and surface degradation of dental alloys [7,16]. Namely, it has been shown by Mellado et al. [7] that fluoride-containing acidic solutions accelerate galvanic corrosion in titanium alloys and CoCr suprastructures.

Oral hygiene products also play a significant role in the corrosion and galvanic interaction of dental alloys. Commercially available mouth rinses and toothpastes may contain components increasing the corrosion and galvanic interaction of dental alloys such as fluoride, chlorhexidine, and ethanol. It has been shown that fluoride-containing products may increase galvanic corrosion of dental alloys [17,18,19,20]. Several experimental studies demonstrated the negative effects of ethanol-based mouth rinses [18,21], Listerine [22,23], and chlorhexidine [24]. It has been shown that alcohol-containing mouthwashes lead to an increased tendency to corrosion in the case of CoCr alloys [21]. The lower susceptibility to corrosion has also been shown for NiTi alloys in alcohol-containing oral antiseptic solution. 

Saliva replacements and other dry mouth products [25] may potentially affect the corrosion of dental alloys. These products may relieve the symptoms of xerostomia [1] for a short period of time [25,26,27], so the patients have to use them many times a day. However, the effect of dry mouth products on the corrosion of dental alloys is poorly investigated. Spirk et al. investigated the electrochemical properties of several saliva replacements and figured out that the electroconductivity of dry mouth products may significantly differ from natural saliva, in the range from 1.06 mS/cm to 8.93 mS/cm [14,28]. Taking into account the lack of research on the effects of essential oral medications and their interaction with various metal dental implants and saliva, in this work, we examined the effects of commercially available fluoride-free dry mouth products on the corrosion and galvanic behavior of the most common dental alloys Ti64 and CoCr. Such studies are necessary to find a suitable and safe oral medication tailored to each patient’s individual characteristics. 

## 2. Materials and Methods

### 2.1. Materials

For the analysis, two different alloys were chosen: Ti-6Al-4V (Ti64) and CoCr. The Ti samples were milled from commercially available Ti grade 5 milling discs. The Co-Cr samples were prepared from purchased Co-Cr dental alloy for metal-ceramic crowns (Co~67%; Cr~27%; Mo~5%; Mn, Si, C < 1%; Techcom, Ulyanovsk, Russia). 

The following commercially available dry mouth products were used as electrolytes: Buccotherm Fresh Breath Spray, Buccotherm Dental Spray, Xerostom Mouth Spray, Aquamed mundspray, GC Dry Mouth, Xerostom gel, and Fusayama Meyer’s artificial saliva (AS). Fusayama Meyer’s artificial saliva (AS) was used as a control solution. The key information related to the studied solutions is presented in Table 1. The pH of the studied solutions was measured using the Starter 3100-F (Ohaus, Shanghai, China) at room temperature (20 °C). 

### 2.2. Sample Preparation

The samples were cut on a Struers Accutom-100 (Struers, Ballerup, Denmark) cutting machine from cylindrical blanks supplied by the manufacturer for the production of dental crowns. Each sample obtained was 5 mm thick. A copper wire was soldered to the upper plane of the samples to provide electrical contact during the sample’s measurements. The samples were then pressed into epoxy resin cylinders using the TechPress 2 (Allied, Lehi, UT, USA) machine with a diameter of 30 mm and a height of 20 mm so that only the bottom surface of the samples was in contact with the environment (or liquid during measurements). The bottom surface area of the CoCr sample was 0.5024 cm^2^, while that of the Ti64 sample was 0.7536 cm^2^. Both specimens were then installed in a MetPrep 3 (Allied, Lehi, UT, USA) polishing machine. Polishing was carried out in several steps with polishing discs with different grit sizes and different solutions. The final polishing solution was 40 nm colloidal silica (OP-U). As a result of this polishing process, the final surface roughness of the sample should be less than Ra = 0.1. After the polishing process, the samples were washed with 99% isopropyl alcohol and dried with compressed air. The prepared samples were used for the study of the electrochemical properties and microstructure characterization. Figure 1 shows a photograph of the samples in the epoxy resin with pre-soldered contacts. After each electrochemical measurement, samples were repolished using the same procedure. 

### 2.3. Electrochemical Characterization

Open circuit potential, impedance spectroscopy, and corrosion characteristics were measured using the Potentiostat/Galvanostat Elins PX40 (Electrochemical Instruments, Chernogolovka, Russia). All measurements were conducted at room temperature (23 °C) with a reversible hydrogen electrode. Impedance spectroscopy was used to determine the conductivity of the solution, with a frequency range from 1 to 500 kHz and an AC signal amplitude of 30 mV. Corrosion tests were performed in the range of −1 to 5 V, with the upper limit of potential being 5 V. The scanning rate was 10 mV s^−1^ for Fusayama Meyer’s artificial saliva (AS), the Buccotherm Dental Spray, Buccotherm Fresh Breath Spray, and Aquamed mundspray.

### 2.4. Surface Morphology Analysis

The scanning electron microscopy (SEM) analysis was conducted in high vacuum using the microscope Quattro S (Thermo Fisher Scientific Inc., Waltham, MA, USA) equipped with EDAX analysis. A semiconductor back reflection electron detector was used to visualize the samples. The accelerating voltage for surface imaging was 10 kV. The working distance to the sample surface was 10 mm. To study the elemental composition, energy dispersive analysis by Bruker XFlash 6|60 was used.

### 2.5. X-ray Diffraction

Ti64 and CoCr alloys before and after electrochemical measurements were analyzed using XRD. Diffractograms were taken between 30° and 90° in Bragg–Brentano geometry, in 2θ-ω scan mode, using CuKα radiation (λ = 1.540605 Å) and at 45 kV and 40 mA. 

## 3. Results and Discussion

### 3.1. Open Circuit Potential Measurements

Before the study of the behavior of the material, the pH of different solutions was studied. The pH levels for all studied solutions are close to 7, within a small variation, as stated in Table 1. After measuring the pH, we conducted experiments on different solutions. The embedded samples in epoxy resin (Figure 1) were positioned at a fixed distance within the studied solutions. 

We used a reversible hydrogen electrode as a reference electrode and measured the open circuit potential between the dental alloy samples once they were placed in the analyzed solutions. Figure 2 illustrates that the open circuit potential of most solutions is within the range of 0.2 to 0.6 V, which is close to that of Fusayama Meyer’s artificial saliva (AS). The primary change in the open circuit potential occurs during the initial few hours, followed by a steady state. Interestingly, Buccotherm Dental Spray demonstrated significantly different results. The obtained result may be explained by different compositions of Buccotherm Dental Spray, which contains only thermal water without Xylitol, preservatives, and other additives. Furthermore, there is a constant change in the potential for Buccotherm Dental Spray, which is attributed to the air dissolved in the solution used for spraying the substance. Over time, the dissolved oxygen in the solution gets displaced, resulting in a constant change in potential. The potential keeps changing for approximately 8 h until it reaches a constant value of approximately −0.7 V. Typically, gel-like solutions separate into distinct phases. For further analysis, we selected Buccotherm (Fresh Breath Spray and Dental Spray) and Aquamed mundspray solutions because they display extreme and average voltage values. Fusayama Meyer’s artificial saliva (AS) was used as a control solution.

### 3.2. Electrochemical Impedance Measurement

Electrochemical impedance spectroscopy (EIS) revealed that solutions exhibit varying resistances. The Nyquist plot, depicting the interrelation of real and imaginary impedance parts, is shown in Figure 3. In order to analyze the data, we employed the equivalent circuit [29] shown on the inset. R1 represents the resistance in the solid phase, namely the internal probe resistance and connecting cable parasitic resistance. The R2 and C2 correspond to the double layer at the solid–liquid interface. The R3 and C3 are the resistance and capacitance of the liquid under research.

Aquamed mundspray demonstrated the lowest resistance value at 240 Ohm, followed by Buccotherm Dental Spray at 460 Ohm, and Fusayama Meyer’s AS at 470 Ohm.

### 3.3. Corrosion Analysis

Corrosion measurements were carried out in four selected solutions. The samples were polished each time according to the procedure described in the experimental part. Measurements in Fusayama Meyer’s AS were carried out both directly with CoCr and Ti64 separately, and together. In joint measurements, one of the materials was used as a reference electrode. The resulting curves are shown in Figure 4a. Corrosion measurements of Ti64 in an artificial environment result in corrosion potential values of −0.66 V and 4.0 × 10^−7^ A, respectively. 

Comparing CoCr and Ti64 in Fusayama Meyer’s AS, it can be concluded that both the corrosion potential and the corrosion current differ. The corrosion potential for CoCr is −0.82 V. In this case, the corrosion potential when measuring samples in pairs is lower than for samples separately. Values obtained for CoCr alloy were replaced with values described in other publications with other solutions [7,21]. There is a difference in the passivation current; however, within the pairs, the passivation currents are the same as the samples outside the pairs. It can be assumed that the presence of a contact pair in the mouth leads to a decrease in the corrosion potential, which indicates the formation of a passivating layer and a more stable pair. The lowest corrosion potential and higher corrosion current for CoCr indicate that the corrosion rate is at the highest. 

Pitting is typically observed in the CoCr alloy at a lower potential. Previous research has shown that the pitting potential value for this alloy is 420 mV. However, in our case, we found that the pitting potential for the alloy is at ~600 mV. The pitting potential for Ti64 is typically higher, as indicated in a recent study [30].

A corrosion study was conducted against glassy carbon for the remaining solutions (Buccotherm Fresh breath, Buccotherm Dental Spray, and Aquamed mundspray). The resulting curves are shown in Figure 4b. Ti64 in Fusayama Meyer’s artificial saliva (AS) had the smallest corrosion potential (indicating the least stability) among the solutions. This behavior may be associated with the presence of a large number of salts in the solution, which reacts with titanium. On the other hand, Ti64 in Aquamed mundspray showed the highest value of −0.03 V. Interestingly, the CoCr alloy in the Aquamed mundspray solution showed a higher value than in AC, indicating its stability in this environment. Similar to Fusayama Meyer’s artificial saliva (AS), pitting formation was observed in the other solutions, and the pitting potential was in the region of 1 V. Table 2 summarizes the obtained data. As seen from Table 2, the corrosion currents are rising in the following series of solutions: Buccotherm Fresh Breath Spray, Buccotherm Dental Spray, and Aquamed mundspray for both dental alloy samples. Fusayama Meyer’s artificial saliva (AS) shows the lowest corrosion potential from the solutions under research for both Ti64 and CoCr samples, which may highlight the highest possible exposure to corrosion in saliva solutions under research. 

### 3.4. Surface Analysis

Before and after studying the corrosion characteristics, a surface analysis was conducted using a scanning electron microscope, as shown in Figure 5. Notably, the formation of additional phases is observed on CoCr and Ti64 surfaces (light areas) even before the electrochemical measurements. On CoCr, these phases follow the crystallization gradient. Significantly, scanning of the region did not reveal any segregation of elements at the grain boundaries. After conducting the corrosion tests, the titanium sample was found to be uniform without any distinctive features. There were rare instances of single defects that could be attributed to pitting corrosion. Meanwhile, the CoCr sample became inhomogeneous. Firstly, the formation of cavities between the grain boundaries is evident. There are also pitting spots. 

Further analysis shows evidence of pitting corrosion and interfacial corrosion, marked with arrows in Figure 6a–c. The pitting corrosion leads to the release and dissolution of cobalt from the surface and remaining surface chromium oxide (Cr^+3^), which is the main substance responsible for the corrosion resistance with a minor contribution of Co and Mo oxides, which can influence the patient’s health [31,32]. Presumably, two anodic peaks in Figure 4b for CoCr alloy may be attributed to the oxidation Mo and Cr, respectively, as was shown in previous studies [33].

The analysis of the sample area using the EDAX analyzer revealed the distribution of elements, as shown in Figure 7. Evidently, during the alloy casting process, molybdenum segregation occurs, which is not visible on polished samples. Previous studies have shown that molybdenum has the lowest solubility in such media [34].

The lattice structure of CoCr alloys can vary depending on the production methods [35,36,37]. SEM images of the samples before corrosion reveal distinct areas corresponding to different phases. However, the XRD analysis of samples before the corrosion only showed one FCC lattice due to a minor quantity of HCP, as shown in Figure 8. Following the corrosion process, distinct peaks appeared in the pattern. According to a previous study, these peaks are characteristic of the HCP lattice [35]. The presence of the second peak may be attributed to the differential etching rates of the HCP and FCC lattices due to packaging density. Conversely, no variations were observed in the XRD pattern of titanium alloys.

## 4. Conclusions

Conducted studies demonstrate the behavior of coupled Ti64 and CoCr alloys in different commercially available dry mouth products compared to Fusayama Meyer’s artificial saliva. The resistance value for Aquamed was close to the natural mixed saliva, while the resistance level of Buccotherm Dental Spray was significantly higher and close to that of Fusayama Meyer’s AS. Additionally, dry mouth products demonstrate different effects on the galvanically coupled Ti64 and CoCr dental alloys. The open circuit potential for Ti64 and CoCr of Buccotherm Dental Spray has a lower value of −0.7 V. The highest value of open circuit potential is possessed by Fusayama Meyer’s solution at ~0.6 V. The values for other solution layers are in the interval between −0.7 and 0.6 V. The corrosion potential of Ti64 is in the range of −0.66 to 0 V, and these alloys did not demonstrate pitting corrosion up to 5 V. The passivation current changed from 4.3 × 10^−6^ to 8.7 × 10^−5^ A cm^−2^. The CoCr alloy demonstrates a different behavior. The corrosion and pitting potentials are lowest in Fusayama Meyer’s artificial saliva (AS) and are equal to −0.82 V and 0.7 V, respectively. In other solutions, the corrosion and pitting potentials shifted, but the value of pitting was close to 1 V. 

It can be argued that the environment of commercially available dry mouth remedies is more favorable for dental alloys in terms of corrosion compared to Fusayama Meyer’s artificial saliva, and continued use of such remedies is not associated with the risk of increased corrosion of metal intraoral devices. However, we did not include fluoride-containing products, which can be more corrosive to dental alloys. The CoCr alloy is more susceptible to corrosion compared to the Ti alloy, so Ti suprastructures are preferable.

## Figures and Tables

**Figure 1 materials-16-04195-f001:**
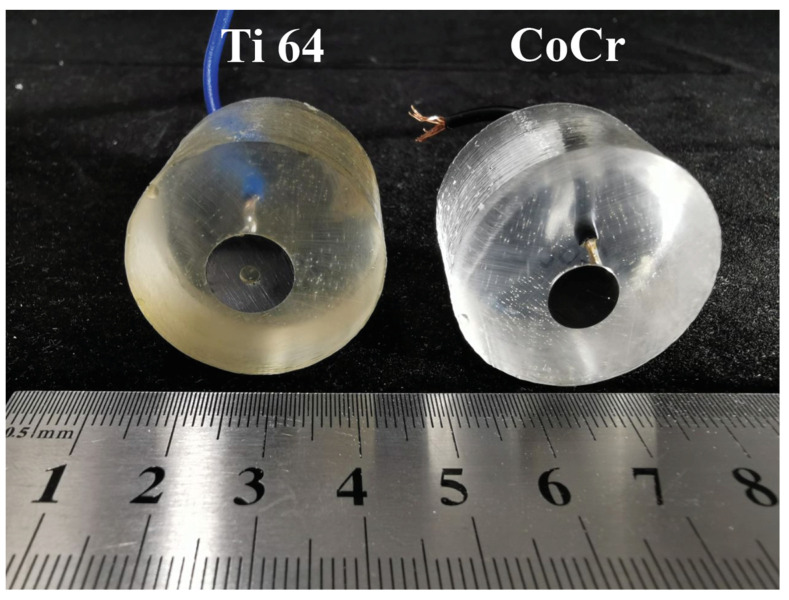
Photo of the dental alloy samples with pre-soldered contacts before the electrochemical measurements.

**Figure 2 materials-16-04195-f002:**
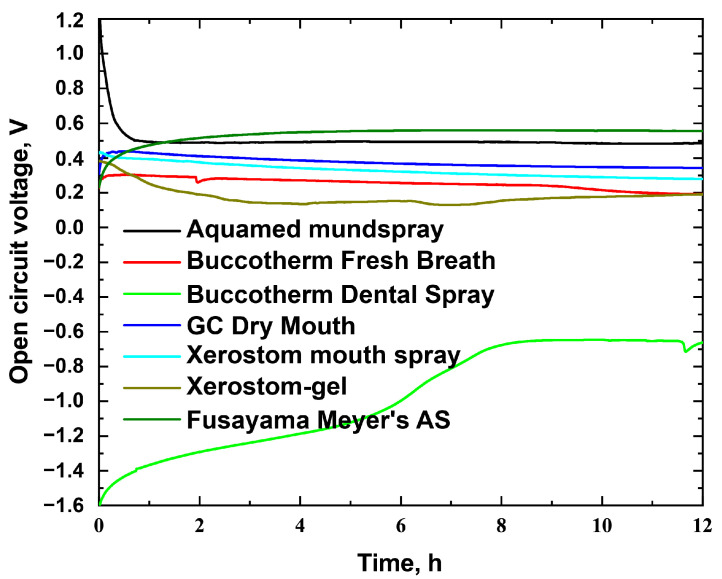
Open circuit voltage of Ti64-CoCr pair in different electrolytes.

**Figure 3 materials-16-04195-f003:**
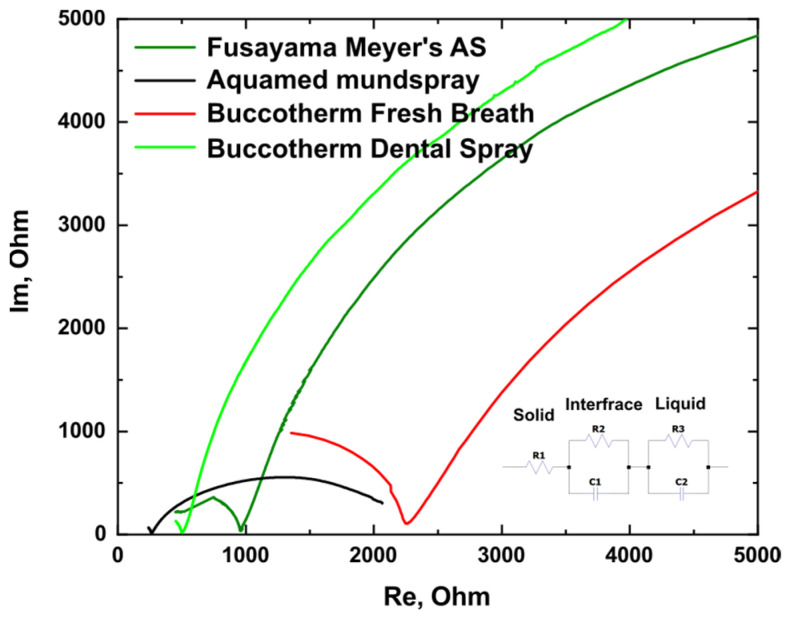
Nyquist plot for the different dry mouth products compared to Fusayama Meyer’s AS.

**Figure 4 materials-16-04195-f004:**
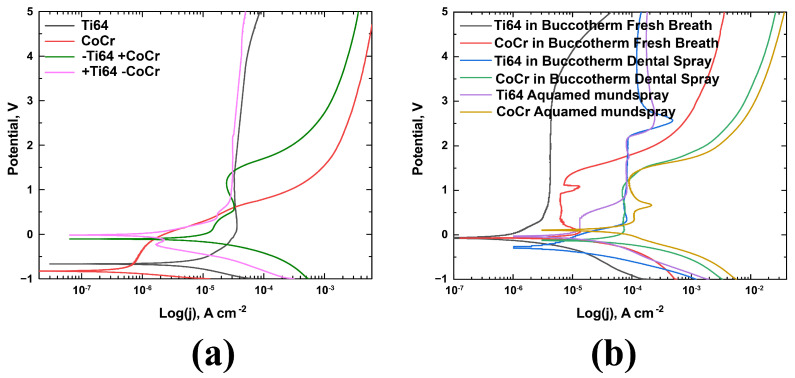
Corrosion curves of CoCr and Ti64 in (**a**) Fusayama Meyer’s AS; (**b**) different electrolytes.

**Figure 5 materials-16-04195-f005:**
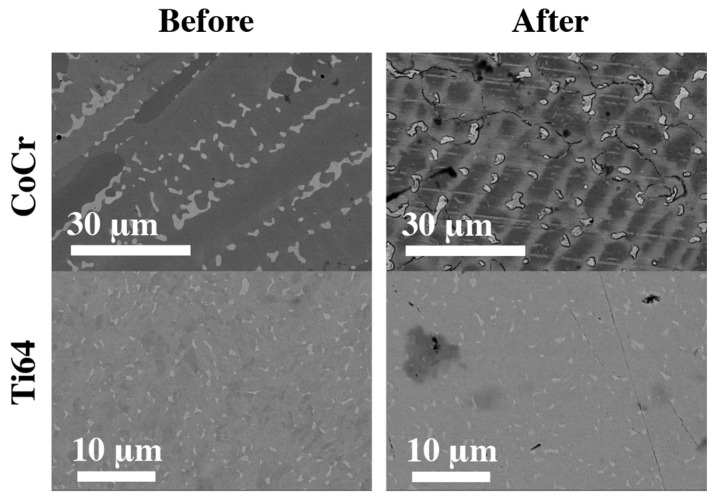
SEM images before and after corrosion in Fusayama Meyer’s artificial saliva (AS).

**Figure 6 materials-16-04195-f006:**
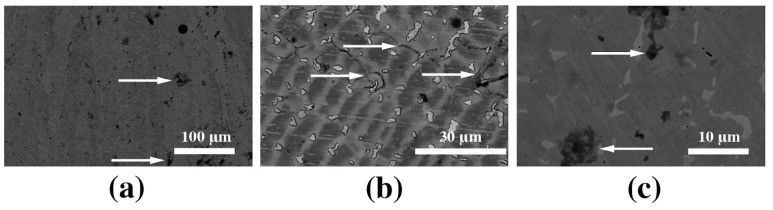
(**a**–**c**) SEM images with different magnifications of the sample after the corrosion test on CoCr. The arrows represent the pitting corrosion and intergrain corrosion.

**Figure 7 materials-16-04195-f007:**
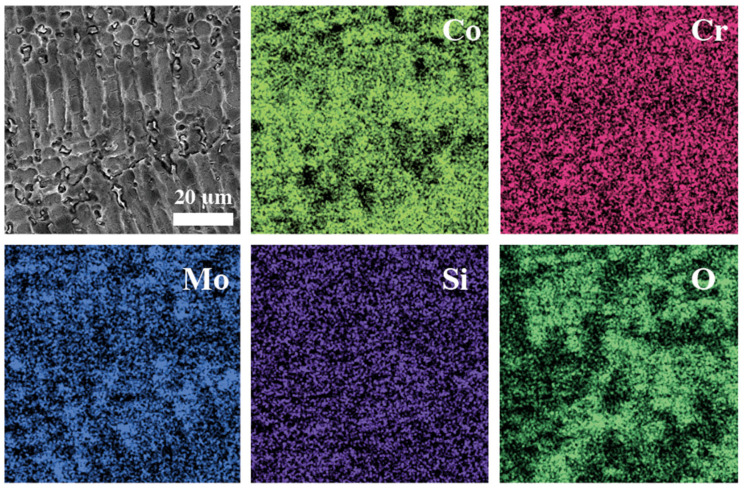
EDAX analysis of CoCr after corrosion tests.

**Figure 8 materials-16-04195-f008:**
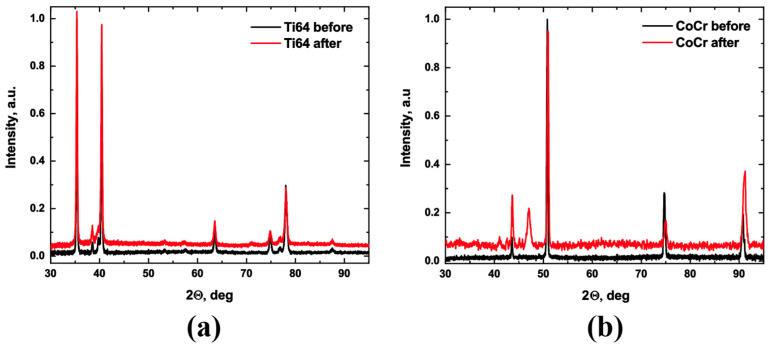
XRD pattern of (**a**) Ti64 and (**b**) CoCr alloys before and after electrochemical measurements.

**Table 1 materials-16-04195-t001:** Key characteristics of the studied solutions.

Studied Solution	Manufacturer	Composition	pH
Fusayama Meyer’s artificial saliva (AS)	The solution was prepared just before the experiment following the formulation	NaCl—0.4 g, KCl—0.4 g, CaCl_2_·2H_2_O—0.8 g, Na_2_HPO_4_—0.7 g, Urea 1.0 g, and distilled water up to 1.0 L	7.0
Buccotherm Fresh Breath Spray	LaboratoireOdost, France	Castera-Verduzan thermal water, Alcohol, Xylitol, Glycerin, Camelia sinesis leaf water, Mentha piperita leaf water, Aroma, Limonene, Benzyl alcohol, and Dehydroacetic acid	7.0
Buccotherm Dental Spray	LaboratoireOdost, France	100% Castéra-Verduzan Thermal Spring water	6.9
Xerostom Mouth Spray	Biocosmetics Laboratories, Spain	Xylitol, Glycerin, Betaine, Panthenol, Carum Petroselinum (Parsley Oil), Calcium Lactate, PEG-40 Hydrogenated Castor Oil, Allantoin, Olea Europaea (Virgin Olive Oil), Tocopheryl Acetate, Aqua, Propylene Glycol, Aroma, D-limonene, Lactic acid, Sodium Methylparaben, Sodium Propylparaben, and Diazolidinyl Urea	7.0
Aquamed mundspray	Hager&Werken GmbH, Germany	Xylitol, Eriodictyon californicum flower/leaf/stem extract, PEG-40 Hydrogenated Castor Oil, Dipotassium phosphate, Lysozyme hydrochloride, Mentha arvensis leaf oil, Magnesium chloride, Calcium chloride, Aqua, Butylene glycol, Aroma, Limonene, Potassium sorbate, Sodium Benzoate, Citric acid, Sodium chloride, and Xanthan gum	6.8
GC Dry Mouth	GC, USA	Diglycerin, Sodium Citrate, Aqua, Aroma, Ethylparaben, Benzyl Alcohol, Cellulose Gum, and Carrageenan	6.86
Xerostom gel	Biocosmetics Laboratories, Spain	Xylitol, Glycerin, Betaine, Panthenol, Potassium Citrate, Potassium phosphate, Calcium Lactate, Tetrapotassium Pyrophosphate, Olea Europaea Fruit Oil (Extra Virgin Olive Oil/Aceite de Oliva Virgen Extra), Tocopheryl Acetate, Aqua, Aroma, Sodium Propylparaben, Sodium Benzoate, Carbomer, and Xanthan Gum	7.03

**Table 2 materials-16-04195-t002:** Corrosion parameters in different solutions at scan rates of 10 mV s^−1^.

Electrode	Solution	E_corr_, V	i_corr_, A cm^−2^	i_pass_, A cm^−2^	E_pitt_, V	i_pitt,_ A cm^2^
Ti64	Fusayama Meyer’s artificial saliva (AS)	−0.66	4.0 × 10^−7^	3.4 × 10^−5^	>5 V	N/A
	Buccoterm Fresh Breath Spray	−0.06	1.2 × 10^−6^	4.3 × 10^−6^	>5 V	N/A
	Buccoterm Dental Spray	−0.26	4.1 × 10^−6^	1.3 × 10^−5^	>5 V	N/A
	Aquamed mundspray	−0.03	6.1 × 10^−6^	8.7 × 10^−5^	>5 V	N/A
CoCr	Fusayama Meyer’s artificial saliva (AS)	−0.82	3.9 × 10^−7^	9.8 × 10^−7^	0.7	2.1 × 10^−5^
	Buccoterm Fresh Breath Spray	−0.07	8.4 × 10^−6^	6.3 × 10^−6^	1.4	1.2 × 10^−5^
	Buccoterm Dental Spray	−0.11	3.3 × 10^−5^	7.1 × 10^−5^	1.5	1.5 × 10^−4^
	Aquamed mundspray	0.11	3.5 × 10^−5^	1.1 × 10^−4^	1.4	1.3 × 10^−4^
-Ti64+CoCr	Fusayama Meyer’s artificial saliva (AS)	−0.11	6.3 × 10^−7^	1.1 × 10^−6^	1.44	1.9 × 10^−6^
+Ti64-CoCr	Fusayama Meyer’s artificial saliva (AS)	−0.01	3.3 × 10^−7^	3.0 × 10^−5^	>5 V	N/A

## Data Availability

All data generated or analyzed during this study are included in this published article.

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
