# Peer review of "The Impact of Commercially Available Dry Mouth Products on the Corrosion Resistance of Common Dental Alloys"

_materials, 2023, doi:10.3390/ma16114195_

Round 1

Reviewer 1 Report

This manuscript contains studies on the influence of selected dry mouth products on the galvanic behavior and corrosion of Ti64 and CoCr dental alloys. The results of these investigations were compared with analogous studies performed in the reference environment of Fusayama Meyer's artificial saliva. In order to realize the goal of this work, the authors used classic experimental methods, which included morphological and electrochemical studies. The results obtained in the investigation might be interesting to researchers working in similar fields.The downside of this work is the lack of comprehensive analysis of the received results on the background of current literature on the subject. A major revision is needed before the work can be further considered for publication. Please find detailed remarks addressing some of the issues below:

 1.The introduction to the work is too general and does not adequately reflect the aim of the research due to the lack of comprehensively performed literature analysis.

 2. Fig. 1 does not add significant scientific value and therefore, removing it from the manuscript is advised.

 3. More information on the operating parameters of the SEM equipmentshould be provided.

 4. There is a lack of comprehensive analysis pertaining to the results of the impedancestudies (subsection 3.2.). Assuming an equivalent circuit model is recommended.

 5. The determined presence of additional phases in the studied samples before and after corrosion should be documented by including the results of phase composition analysis in the work that were performed by XRD.

 6. The two literature references included in the conclusions should be located in the section where the experimental results are discussed.

 7. The conclusions should be shortened, because they are too elaborate.

The language could be improved to some degree.

Author Response

Dear Reviewer,

First of all, we want to thank you for spending your time to provide a review of our paper. We have considered all comments carefully and made several modifications in our manuscript accordingly. We sincerely hope the revised manuscript suits for the publication in Materials.

The detailed response is provided below. In our response, your comments are highlighted in bold. We provide the revised manuscript, where the changes in the text addressing comments by the Reviewers are highlighted in yellow. We refer to the specific pages in the revised manuscript materials in our response.

Reviewer 1.

Comment №1. The introduction to the work is too general and does not adequately reflect the aim of the research due to the lack of comprehensively performed literature analysis.

Author response: We thank you for this comment. We have added more specific data from the existing research to the introduction part on pages 1-2 in order to reflect the aim of the research.

Comment №2. Fig. 1 does not add significant scientific value and therefore, removing it from the manuscript is advised.

Author response: Thank you for your comment. We would like to keep Figure 1 as a visualization of the prepared specimens, their sizes and shapes for a wide range of readers from different fields.

Comment №3 More information on the operating parameters of the SEM equipment should be provided.

Author response: Thank you for your comment. We have added operating parameters information in section 2.4. Surface morphology analysis on page 4.

Comment #4. There is a lack of comprehensive analysis pertaining to the results of the impedance studies (subsection 3.2.). Assuming an equivalent circuit model is recommended.

Author response: Thank you for this comment. We agree that there is a lack of information about results of impedance analysis. We have added information about the equivalent circuit used in the analysis in Figure 3 on page 6 and determined the resistance of the fluid under study.

Comment #5. The determined presence of additional phases in the studied samples before and after corrosion should be documented by including the results of phase composition analysis in the work that were performed by XRD.

Author response: We have carried out XRD experiment in order to identified the presence of additional phases. XRD pattern for the studied samples before and after corrosion and related discussion are shown in Figure 8 in the text on pages 9-10.

Comment #6. The two literature references included in the conclusions should be located in the section where the experimental results are discussed.

Author response: We agree that literature references and discussion in the conclusions should be moved in section results and discussion.

Comment #7. The conclusions should be shortened, because they are too elaborate.

Author response: We rewrote the conclusion on page 10, omitting the details described in the results section.

Reviewer 2 Report

Thank you for allowing me to review the manuscript whose purpose was to investigate the behavior of common titanium and cobalt-chromium alloys in interaction with various dry mouth products

The topic is interesting, but corrections are necessary before publication.

1. Introduction in abstract part is long. Please shorten it.

2. The methodology is not specified in the abstract. Please present the results in such a way that they are supported by numbers, not only descriptively.

3. State the conclusion in the abstract, leave the narrative work for yourself when the conclusion can be longer. One concluding sentence.

4. Methodology - what was the total size of samples. In what form are the alloys prepared, what dimension? The methodology is not detailed! More precisely, it does not describe at all how the research was carried out, only with what material, means and methods, but it is not clarified how, and I would be interested.

5. The obtained results are not sufficiently clarified and compared with other studies.

6. The literature is not written in accordance with the instructions of the Journal.

Author Response

Dear Reviewer,

First of all, we want to thank you for spending your time to provide a review of our paper. We have considered all comments carefully and made several modifications in our manuscript accordingly. We sincerely hope the revised manuscript suits for the publication in Materials.

The detailed response is provided below. In our response, your comments are highlighted in bold. We provide the revised manuscript, where the changes in the text addressing comments by the Reviewers are highlighted in yellow. We refer to the specific pages in the revised manuscript materials in our response.

Reviewer 2.

Comment #1. Introduction in abstract part is long. Please shorten it.

Author response: Thank you for your advice. We have shortened introductory part in abstract on page 1.

Comment #2. The methodology is not specified in the abstract. Please present the results in such a way that they are supported by numbers, not only descriptively.

Author response: Thank you for your comment. We have added method and results in abstract on page 1.

Comment #3. State the conclusion in the abstract, leave the narrative work for yourself when the conclusion can be longer. One concluding sentence.

Author response: Thank you for your advice. We have corrected the abstract and added conclusion statement on page 1.

Comment #4. Methodology - what was the total size of samples. In what form are the alloys prepared, what dimension? The methodology is not detailed! More precisely, it does not describe at all how the research was carried out, only with what material, means and methods, but it is not clarified how, and I would be interested.

Author response: Thank you for pointing this out. We agree that there is absence of details about sample preparation. The samples were cut on a Struers Accutom-100 cutting machine from cylindrical blanks supplied by the manufacturer for the production of dental crowns. Each sample obtained was 5 mm thick. A copper wire was soldered to the upper plane of the samples to provide electrical contact during the sample’s measurements. The samples were then pressed into epoxy resin cylinders using the TechPress 2 machine with a diameter of 30 mm and a height of 20 mm so that only the bottom surface of the samples was in contact with the environment (or liquid during measurements). The bottom surface area of the CoCr sample was 0.5024 cm2, while that of the Ti64 sample was 0.7536 cm2. Both specimens were then installed in a MetPrep 3 polishing machine. Polishing was carried out in several steps with polishing discs with different grit sizes and different solutions. The final polishing solution was 40 nm colloidal silica (OP-U). As a result of this polishing process, the final surface roughness of the sample should be less than Ra=0.1. After the polishing process, the samples were washed with 99% isopropyl alcohol and dried with compressed air. The prepared samples were used for the study of the electrochemical properties and microstructure characterization. Figure. 1 shows a photograph of the samples in the epoxy resin with pre-soldered contacts. After each electrochemical measurement samples were repolished using the same procedure.

We also added this information in the manuscript in section 2.2 Sample preparation on page 4.

Comment #5. The obtained results are not sufficiently clarified and compared with other studies.

Author response: Thank you for pointing that out. We have rewritten the introduction part and expanded the discussion of the research already done in the area of investigating the effects of hygiene products on the corrosion of metal implants on pages 1-2. We also pointed out that there is a lack of research on the effect of dry mouth medications on the corrosion of the implants used, which justifies the choice of our study objects.

Comment #6. The literature is not written in accordance with the instructions of the Journal.

Author response: We have updated the literature sources in accordance with the instructions of the Materials Journal.

Reviewer 3 Report

This study seems meaningful as a study on galvanic corrosion of dental metal materials. However, it can be published after major revision.

1.      English needs to be corrected.

2.      In the experimental method, when conducting a corrosion test on a specimen, the degree of polishing of the surface must be presented, and conditions such as deoxygenation of the solution, pH, and temperature of the solution are required. In addition, please describe whether the experiment was conducted according to the ASTM method or the TC106 method. If the polarization curve is obtained by performing up to 5V, it is difficult to distinguish whether it is corrosion by pitting or intergranular corrosion in the corroded matrix, so most of the tests are performed up to 2V and evaluated. This is because it is similar to the clinical environment. In the preparation of the alloy specimen, if you look at the after-corrosion morphology, the dendritic structure that appears in the specimen after casting appears in the CoCr alloy. In the case of commercial sales, it is provided after homogenization treatment. Please suggest if you have used the specimen after casting.

3.     Corrosion experiments were conducted in 7 solutions. Please present the mechanism along with the discussion of the main anions involved in galvanic corrosion in each solution. In addition, it is necessary to measure the conductivity of the electrolyte, and the corrosion rate is influenced accordingly. In Figure 3, please add the data obtained from the 7 solutions to the EIS results.

4.      In Figure 4, on the polarization curve, a second anodic peak and reaction is shown, especially in CoCr, and it is necessary to explain the phenomenon.

5.     Why are the potential scan rates(1, 10) different in Table 2? This is because it is difficult to compare polarization curves when potential scan rates are different.

6.     In Figure 5, the phases appearing in CoCr alloy and Ti alloy are identified by XRD analysis and displayed.

7.      The title of the thesis is Galvanic Corrosion, but since there is no galvanic test data, it must be presented and explained. Present the current change with time between two metals at the same time to suggest which metal is corroded.

 English needs to be corrected.

Author Response

Dear Reviewer,

First of all, we want to thank you for spending your time to provide a review of our paper. We have considered all comments carefully and made several modifications in our manuscript accordingly. We sincerely hope the revised manuscript suits for the publication in Materials.

The detailed response is provided in the attachment. In our response, your comments are highlighted in bold. We provide the revised manuscript, where the changes in the text addressing comments by the Reviewers are highlighted in yellow. We refer to the specific pages in the revised manuscript materials in our response.

Round 2

Reviewer 1 Report

In my opinion, the authors have addressed all reviewers' comments in the revised version of the paper. Accordingly, I recommend the paper for publication in its present form.

Reviewer 2 Report

Thanks for accepting requested modifications. 

Your manuscript now seem much better. 

Reviewer 3 Report

This MS was well revised according to reviewer's comments. Therefore, reviewer recommends this MS for publication in this Journal.